# Modelling locust foraging: How and why food affects group formation

**Fillipe Georgiou**[1]*, **Camille Buhl**[2], **J. E. F. Green**[3], **Bishnu Lamichhane**[1], **Ngamta Thamwattana**[1]

**1** School of Mathematical and Physical Sciences, University of Newcastle, Callaghan, Australia, **2** School of Agriculture, Food and Wine, University of Adelaide, Adelaide, Australia, **3** School of Mathematical Sciences, University of Adelaide, Adelaide, Australia

* fillipe.georgiou@uon.edu.au

## Abstract

Locusts are short horned grasshoppers that exhibit two behaviour types depending on their local population density. These are: solitarious, where they will actively avoid other locusts, and gregarious where they will seek them out. It is in this gregarious state that locusts can form massive and destructive flying swarms or plagues. However, these swarms are usually preceded by the aggregation of juvenile wingless locust nymphs. In this paper we attempt to understand how the distribution of food resources affect the group formation process. We do this by introducing a multi-population partial differential equation model that includes non-local locust interactions, local locust and food interactions, and gregarisation. Our results suggest that, food acts to increase the maximum density of locust groups, lowers the percentage of the population that needs to be gregarious for group formation, and decreases both the required density of locusts and time for group formation around an optimal food width. Finally, by looking at foraging efficiency within the numerical experiments we find that there exists a foraging advantage to being gregarious.

**Data Availability Statement:** All relevant data are within the manuscript and its Supporting information files.

**Funding:** JEFG received support from the School of Mathematical Sciences and the Faculty of

## Author summary

Locusts are short horned grass hoppers that live in two diametrically opposed behavioural states. In the first, solitarious, they will actively avoid other locusts, whereas the second, gregarious, they will actively seek them out. It is in this gregarious state that locusts form the recognisable and destructive flying adult swarms. However, prior to swarm formation juvenile flightless locusts will form marching hopper bands and make their way from food source to food source. Predicting where these hopper bands might form is key to controlling locust outbreaks. Research has shown that changes in food distributions can affect the transition from solitarious to gregarious. In this paper we construct a mathematical model of locust-locust and locust-food interactions to investigate how food distributions affect the aggregation of juvenile locusts, termed groups, an important precursor to hopper bands. Our findings suggest that there is an optimal food distribution for group formation and that being gregarious increases a locusts ability to forage when food becomes more patchy.

Engineering, Computer and Mathematical Sciences, University of Adelaide through the Special Studies Programme between January-July 2019 (during which time this work was initiated). FG and NT were supported by the University of Newcastle, Australia, via a RTP PhD scholarship and start-up support respectively. The funders had no role in study design, data collection and analysis, decision to publish, or preparation of the manuscript.

**Competing interests:** The authors have declared that no competing interests exist.

## Introduction

Locust swarms have plagued mankind for millennia, affecting every continent except Antarctica and impacting on the lives of 1 in 10 people [1]. A single locust swarm can contain millions of individuals [2] and in a single day is able to move up to 200 kilometres [3]; with each locust being able to consume its own body weight in food [4]. Locusts have played a role in severe famine [5], disease outbreaks [6], and even the toppling of dynasties [7]. More recently, in March 2020 a perfect storm of events caused the worst locust outbreaks in over 25 years in Ethiopia, Somalia and Kenya during the COVID-19 pandemic [8]. Damaging tens of thousands of hectares of croplands and pasture, these outbreaks presented an unprecedented threat to food security and livelihoods in the Horn of Africa. In addition, the onset of the rainy season meant the locusts were able to breed in vast numbers raising the possibility of further outbreaks [9].

Locusts are short horned grasshoppers that exhibit density-dependent phase-polyphenism, i.e., two or more distinct phenotype expressions from a single genotype depending on local population density [10]. In locusts there are two key distinct phenotypes, solitarious and gregarious, with the process of transition called gregarisation. Gregarisation affects many aspects of locust morphology from colouration [11], to reproductive features [12], to behaviour [13]. Behaviourally, solitarious locusts are characterised by an active avoidance of other locusts, whereas gregarious locusts are strongly attracted to other locusts. Gregarisation is brought about by locusts crowding together and can be reversed by isolating the individuals [4]. In the Desert locust (*Schistocerca gregaria*), gregarisation can take as little as 4 hours with the timeframe for reversal dependant on the length of time the individual has been gregarious (again, potentially as little as 4 hours) [10].

In this gregarious state there is greater predator avoidance on the individual level [14], the group display of aposematic colours has a greater effect of predator deterrence [15], and the resulting aggregations may act as a means of preventing mass disease transmission [16]. It is also in the gregarious state that locusts exhibit large scale and destructive group dynamics with flying swarms of adult locusts being perhaps the most infamous manifestation of this.

Despite the destruction caused by adult swarms, the most crucial phase for locust outbreak detection and control occurs when wingless nymphs form hopper bands, large groups of up to millions of individuals marching in unison [4]. Depending on the species, these groups may adopt frontal or columnar formations, the former being comet like in appearance with dense front and less dense tail [17], and the latter being a network of dense streams [4]. As a precursor to hopper bands, nymphs will form gregarious aggregations or groups, i.e. a large mass of gregarious nymphs. Understanding the group dynamics of gregarious locusts are key to improving locust surveys and control by increasing our ability to understand and predict movement.

In addition to the group dynamics, better knowledge of locust interactions with the environment would help to improve the prediction of outbreaks [18]. On longer time-scales, environmental conditions such as rain events synchronize locust lifecycles and can lead to repeated outbreaks [10]. On shorter time-scales, changes in resource distributions at both small and large spatial scales have an effect on locust gregarisation [19–22]. It is these short time-scale locust-environment interactions that we investigate in this paper, using mathematical modelling to further understand both their effect on group formation and if there is any advantage to gregarisation in this context.

As all the mentioned behaviours arise from simple inter-individual interactions, understanding the group dynamics of gregarious locusts can also give deep insight into the underlying mechanisms of collective behaviour. Consequently they are an important subject of

mathematical modelling efforts. Self propelled particle (SPP) models are a frequently used approach in which locusts are modelled as discrete individuals who update their velocity according to simple interaction rules. SPPs are catergorised as second order models if they include particle inertia, and first order (or kinematic) if inertia is neglected [23]. While second order SSP models have been used fairly extensively as they able to capture collective movement mechanisms such as alignment or pursuit/escape interactions [2, 17, 24, 25]; first order SPP models are still useful for modelling the more disordered stages of locust behaviour [26, 27]. One downside of SPP models is that there are few analytical tools available to study their behaviour. In contrast, continuum models, in which locusts are represented as a population density that is a function of space and time, can be analysed using an array of tools from the theory of partial differential equations (PDEs). They are most appropriately employed when there are a large number of individuals since they do not account for individual behaviour, instead giving a representation of the average behaviour of the group. The latter (continuum) approach is adopted in this paper.

The non-local aggregation equation, first proposed by Mogilner and Edelstein-Keshet [28], is a common continuum PDE analogue of the kinematic SPP model [29, 30]. It is a conservation of mass equation of the form

$$\frac{\partial \rho}{\partial t} + \nabla \cdot [(-\nabla Q \star \rho)\rho] = 0,$$

where $Q$ is defined as some social interaction potential, $\rho$ is the density (either mass or population per unit area) of the species in question, and $\star$ is the convolution operation. For this type of model, the existence and stability of swarms has been proven [28], and both travelling wave solutions [28] and analytic expressions for the steady states [27] have been found. This model has been further extended to include non-linear local repulsion which leads to compact and bounded solutions [31]. Usually used for single populations, the model has been further adapted to consider multiple interacting populations [32]. While the kinematic model does not capture complex behaviours such as alignment, the steady state solutions determine the spatial shape and density of flock solutions of second order models (i.e. collective movement of individuals in the same direction) [23, 33].

In a 2012 paper, Topaz et. al. [34] used a multi-population aggregation equation to model locusts as two distinct behavioural sub-populations, solitarious and gregarious. By considering the locust-locust interactions and the transition between the two states, they were able to deduce both the critical density ratio of gregarious locusts that would cause a group to form and visualised the rapid transition once this density ratio had been reached [34]. Similarly, in our work we focus on the formation of aggregations (or groups) of gregarious locusts, visualised as a clump of gregarious locusts, rather than on the collective movement dynamics in hopper bands. For simplicity, the Topaz model focused on inter-locust interactions and ignored interactions between locusts and the environment. While there exists some continuum models of locust food interactions to investigate the effect of food on peak locust density [35] or to consider hopper band movement [36], we are not aware of any studies that consider locust-locust and locust-food interactions as well as gregarisation in a continuum setting.

The aims of this paper are threefold. Firstly, to introduce a new mathematical model that extends the 2012 Topaz model by including both locust-food dynamics and local repulsion. The model is based on an idealised locust which has both long and short range locust interactions and only interacts with food when it come into direct contact with it. Secondly, we use our new model to investigate how the spacial distribution of food affects the gregarisation and group formation process. Finally, we consider under what conditions being gregarious might confer an advantage compared to being solitarious in terms of access to food.

This paper is organised as follows: we begin with the derivation of a PDE model based on our idealised locust. Then, we look at some mathematical properties of our model with a homogeneous food distribution. We next use numerical simulations to investigate the effect of food distribution on group formation, and the relative foraging advantage of gregarisation. Finally, we summarise our results and offer ideas for further exploration of the model.

## Models and methods

In this section we present a PDE model of locust foraging that includes both local inter-individual and food interactions and non-local inter-individual interactions. In order to simplify the model we make the following assumptions about locust behaviour. 1). Locusts can be classified as either solitarious or gregarious. 2). Locusts only interact with food resources when they come into direct contact with them. 3). Local interactions between locusts (both gregarious and solitarious) are repulsive (i.e. they avoid close physical contact). 4). Solitarious locusts experience a non-local (i.e. longer-ranged) repulsion from other locusts of either type. 5). Gregarious locusts experience a non-local long-range attraction and short-range repulsion from other locusts, which is consistent with them forming a well-spaced aggregation [25]. 6). The nature (attractive or repulsive) and strengths of all interactions are constant in time.

### Model derivation

In this model locusts are represented as a density of individuals (number per unit area) in space and time and are either solitarious, $s(\boldsymbol{x}, t)$, or gregarious $g(\boldsymbol{x}, t)$, with the total local density defined as $\rho(\boldsymbol{x}, t) = s(\boldsymbol{x}, t) + g(\boldsymbol{x}, t)$. For later convenience we will also define the total mass of locusts as

$$M = \int \rho(\boldsymbol{x}, t)\, d\boldsymbol{x} \tag{1}$$

and the global gregarious mass fraction as

$$\phi_g(t) = \frac{\int g(\boldsymbol{x}, t)\, d\boldsymbol{x}}{M}. \tag{2}$$

We assume that the time-scale of gregarisation is shorter than the life cycle of locusts, ignoring births and deaths and thus conserving the total number of locusts. We allow for a transition from solitarious to gregarious and vice-versa depending on the local population density. Hence, conservation laws give equations of the form

$$\frac{\partial g}{\partial t} + \nabla \cdot \left( \boldsymbol{J}_{g_{\text{local}}} + \boldsymbol{J}_{g_{\text{non-local}}} \right) = K(s, g), \tag{3a}$$

$$\frac{\partial s}{\partial t} + \nabla \cdot \left( \boldsymbol{J}_{s_{\text{local}}} + \boldsymbol{J}_{s_{\text{non-local}}} \right) = -K(s, g), \tag{3b}$$

where $\boldsymbol{J}_{(s,g)_{\text{local}}}$ is the flux due to local interactions, $\boldsymbol{J}_{(s,g)_{\text{non-local}}}$ is the flux due to non-local interactions, and $K(s, g)$ represents the transition between the solitarious and gregarious states.

In addition to locust densities, we include food resources in our model and let $c(\boldsymbol{x}, t)$ denote the food density (mass of edible material per unit area). We assume that locust food consumption follows the law of mass action and on the time-scale of group formation food production

is negligible, giving

$$\frac{\partial c}{\partial t} = -\kappa c(\boldsymbol{x}, t)\rho(\boldsymbol{x}, t), \tag{4}$$

where $\kappa$ is the locust's food consumption rate.

**Local interactions.**   We now turn to specifying the local interaction terms in Eq (3a) and (3b). These are captured by taking the continuum limit of a lattice model (this should, however, be only considered an asymptotic approximation [37, 38]) following the work of Painter and Sherratt [39]. We begin by considering solitarious locust movement on a one-dimensional lattice with spacing $\Delta x$ (we assume that local gregarious locust behaviour is the same resulting in a similar derivation). Let $s_i^t$ be the number of solitarious locusts at site $i$ at time $t$, and let $g_i^t$, $\rho_i^t$, and $c_i^t$ be similarly defined.

We assume that the transition probability for a locust at the $i^{th}$ site depends on the food density at that site, and the relative population density between the current site and neighbouring sites. If we let $\mathcal{T}_i^{\pm}$ be the probability at which locusts at site $i$ move to the right, +, and left, −, during a timestep, then our transition probabilities are

$$\mathcal{T}_i^{\pm} = F(c_i)(\alpha + \beta(\tau(\rho_i) - \tau(\rho_{i\pm1}))),$$

where $F$ is a function of food density, $\tau$ is a function related to the local locust density, and $\alpha$ and $\beta$ are constants. If nutrients are abundant at the current site, then we assume locusts are less likely to move to a neighbouring site, which implies $F$ is a decreasing function. We set,

$$F(c_i) = e^{-\frac{c_i}{c_0}},$$

where $c_0$ is a constant related to how long a locust remains stationary while feeding. We further assume that as the locust population density rises at neighbouring sites relative to the population density of the current site, the probability of moving to those sites decreases proportional to the number of collisions between individuals that would occur. Using the law of mass action, this gives,

$$\tau(\rho) = \rho^2,$$

with any constant of proportionality being subsumed into $\beta$. Thus, our transition probabilities are

$$\mathcal{T}_i^{\pm} = e^{-\frac{c_i}{c_0}}(\alpha + \beta(\rho_i^2 - \rho_{i\pm1}^2)).$$

Then the number of individuals in cell $i$ at time $t + \Delta t$ is given by

$$s_i^{t+\Delta t} = s_i^t + \mathcal{T}_{i+1}^- s_{i+1}^t + \mathcal{T}_{i-1}^+ s_{i-1}^t - (\mathcal{T}_i^- + \mathcal{T}_i^+)s_i^t.$$

From this, we can deduce the continuum limit for both solitarious and gregarious locust densities, and find our local flux as

$$J_{g_{\text{local}}} = -D\left[\frac{\partial}{\partial x}\left(ge^{-\frac{c}{c_0}}\right) + \gamma g\rho e^{-\frac{c}{c_0}}\frac{\partial\rho}{\partial x}\right],$$

$$J_{s_{\text{local}}} = -D\left[\frac{\partial}{\partial x}\left(se^{-\frac{c}{c_0}}\right) + \gamma s\rho e^{-\frac{c}{c_0}}\frac{\partial\rho}{\partial x}\right],$$

where $D$ and $\gamma$ are continuum constants related to $\alpha$ and $\beta$ respectively (and the number of dimensions, for a full derivation see S1 Appendix). In higher dimensions, the expressions for

the fluxes are:

$$\boldsymbol{J}_{g_{\text{local}}} = -D\left[\nabla\left(ge^{-\frac{c}{c_0}}\right) + \gamma g\rho e^{-\frac{c}{c_0}}\nabla\rho\right], \tag{5a}$$

$$\boldsymbol{J}_{s_{\text{local}}} = -D\left[\nabla\left(se^{-\frac{c}{c_0}}\right) + \gamma s\rho e^{-\frac{c}{c_0}}\nabla\rho\right]. \tag{5b}$$

**Non-local interactions.** For our non-local interactions, we adopt the fluxes used by Topaz et. al. [34]. By considering each locust subpopulation, solitarious and gregarious, as having different social potentials, we obtain the following expressions for the non-local flux

$$\boldsymbol{J}_{g_{\text{non-local}}} = -\nabla(Q_g \star \rho)g, \tag{6a}$$

$$\boldsymbol{J}_{s_{\text{non-local}}} = -\nabla(Q_s \star \rho)s. \tag{6b}$$

We also adopt the functional forms of the social potentials used by Topaz et. al. [34], as they are used extensively in modelling collective behaviour and are well studied [27]. They are based on the assumption that solitarious locusts have a long range repulsive social potential and gregarious locusts have a long range attractive and a shorter range repulsive social potential. The social potentials are given by,

$$Q_s(\boldsymbol{x}) = R_s e^{\frac{-|\boldsymbol{x}|}{r_s}} \text{ and } Q_g(\boldsymbol{x}) = R_g e^{\frac{-|\boldsymbol{x}|}{r_g}} - A_g e^{\frac{-|\boldsymbol{x}|}{a_g}},$$

where, $R_s$ and $r_s$ are the solitarious repulsion strength and sensing distance respectively. Similarly, $R_g$ and $r_g$ are the gregarious repulsion strength and sensing distance. Finally $A_g$ and $a_g$ are the gregarious attraction strength and sensing distance.

**Gregarisation dynamics.** For the rates at which locusts become gregarious (or solitarious) we again follow the work of Topaz et. al. [34]. We assume that solitarious locusts transition to gregarious is a function of the local locust density (and vice versa). This gives our equations for kinetics as,

$$K(s,g) = -f_1(\rho)g + f_2(\rho)s, \tag{7}$$

where, $f_1(\rho)$ and $f_2(\rho)$ are positive functions representing density dependant transition rates. To make our results more directly comparable we again use the same functional forms as Topaz et. al. [34]:

$$f_1(\rho) = \frac{\delta_1}{1 + \left(\frac{\rho}{k_1}\right)^2}, \tag{8a}$$

$$f_2(\rho) = \frac{\delta_2\left(\frac{\rho}{k_2}\right)^2}{1 + \left(\frac{\rho}{k_2}\right)^2}, \tag{8b}$$

where $\delta_{1,2}$ are maximal phase transition rates and $k_{1,2}$ are the locust densities at which half this maximal transition rate occurs.

**A system of equations for locust gregarisation including food interactions.** By substituting our flux expressions, (5a) through to (6b), and kinetics term (7), into our conservation equations, (3a) and (3b), and rearranging the equation into a advection diffusion system, we

obtain the following system of equations

$$\frac{\partial g}{\partial t} + \nabla \cdot (g\boldsymbol{v}_g) - D\nabla \cdot \left[ e^{-\frac{c}{c_0}}\nabla g \right] \quad = -f_1(\rho)g + f_2(\rho)s, \tag{9a}$$

$$\frac{\partial s}{\partial t} + \nabla \cdot (s\boldsymbol{v}_s) - D\nabla \cdot \left[ e^{-\frac{c}{c_0}}\nabla s \right] \quad = f_1(\rho)g - f_2(\rho)s, \tag{9b}$$

$$\frac{\partial c}{\partial t} \quad = -\kappa c(\boldsymbol{x}, t)\rho(\boldsymbol{x}, t). \tag{9c}$$

with

$$\boldsymbol{v}_g = -\nabla(Q_g \star \rho) + De^{-\frac{c}{c_0}}\left( \frac{1}{c_0}\nabla c - \gamma\rho\nabla\rho \right),$$

and

$$\boldsymbol{v}_s = -\nabla(Q_s \star \rho) + De^{-\frac{c}{c_0}}\left( \frac{1}{c_0}\nabla c - \gamma\rho\nabla\rho \right),$$

where $f_1$, $f_2$, $Q_s$, and $Q_g$ are previously defined.

**Non-dimensionalisation.** We non-dimensionalise Eq (9a), (9b) and (9c), and the explicit expressions for $f_1$, $f_2$, $Q_s$, and $Q_g$, using the following scalings

$$t = \frac{1}{\delta_2}\bar{t}, \ \boldsymbol{x} = a_g\bar{\boldsymbol{x}}, \ (\rho, s, g) = k_1(\bar{\rho}, \bar{s}, \bar{g}), \ \text{and} \ c = c_0\bar{c}.$$

Then, dropping the bar notation, the dimensionless governing equations are

$$\frac{\partial g}{\partial t} + \nabla \cdot (g\boldsymbol{v}_g) - D^*\nabla \cdot [e^{-c}\nabla g] \quad = -f_1^*(\rho)g + f_2^*(\rho)s, \tag{10a}$$

$$\frac{\partial s}{\partial t} + \nabla \cdot (s\boldsymbol{v}_s) - D^*\nabla \cdot [e^{-c}\nabla s] \quad = f_1^*(\rho)g - f_2^*(\rho)s, \tag{10b}$$

$$\frac{\partial c}{\partial t} \quad = -\kappa^* c(x, t)\rho(x, t), \tag{10c}$$

where

$$\boldsymbol{v}_g = -\nabla Q_g^* \star \rho + D^*e^{-c}(\nabla c - \gamma^*\rho\nabla\rho), \ \boldsymbol{v}_s = -\nabla Q_s^* \star \rho + D^*e^{-c}(\nabla c - \gamma^*\rho\nabla\rho),$$

and

$$Q_g^* = R_g^* e^{\frac{-|x|}{r_g^*}} - A_g^* e^{-|x|}, \ Q_s^* = R_s^* e^{\frac{-|x|}{r_s^*}},$$

$$f_1^*(\rho) = \frac{\delta^*}{1 + \rho^2}, \ f_2^*(\rho) = \frac{(\rho k)^2}{1 + (\rho k)^2}.$$

Note that we have introduced the following dimensionless parameters,

$$D^* = \frac{D}{\delta_2 a_g^2}, \ k = \frac{k_1}{k_2}, \ \delta^* = \frac{\delta_1}{\delta_2}, \ \gamma^* = k_1^2 \gamma, \ \kappa^* = \frac{\kappa k_1}{\delta_2},$$

$$R_g^* = \frac{R_g k_1}{\delta_2 a_g}, \ A_g^* = \frac{A_g k_1}{\delta_2 a_g}, \ R_s^* = \frac{R_s k_1}{\delta_2 a_g}, \ r_g^* = \frac{r_g}{a_g}, \ r_s^* = \frac{r_s}{a_g}.$$

For notational simplicity we drop the $\cdot^*$ notation in the rest of the paper.

## Results

### PDE model analysis

In this section we investigate the behaviour of our model with a spatially uniform and temporally constant food density. This assumption corresponds to environments where the lengthscale of the food footprint is larger than the lengthscale over which the locusts are distributed, and where the rate of food consumption is negligible compared to the speed of locust interactions. Aside from simplifying the analysis, this assumption also provides a baseline with which to compare our later results, and hence assess the impact of a patchy food distribution. Using this and other simplifying assumptions, we are able to calculate the maximum density and size of gregarious groups for both small and large numbers of locusts. We then consider the linear stability of the homogeneous steady states to investigate how the availability of food affects group formation, before finally investigating how the center of mass is affected by locust interactions. Full details of all calculations can be found in S2 Appendix.

**Density of gregarious groups.** Under some simplifying assumptions we can estimate the maximum density and width of gregarious locusts for both small and large numbers of locusts (i.e. as $M \to 0$ and $M \to \infty$, respectively), termed the small and large mass limits, in one dimension. To begin, we assume that $c$ is constant and not depleting, there are minimal solitarious locusts present in the group (i.e. $\rho \approx g$), and the effect of phase transitions in the group is negligible (i.e. $f_1(\rho)s = f_2(\rho)g = 0$). Finally, while the support of $g$ is infinite (due to the linear diffusion) the bulk of the mass is contained as a series of aggregations; consequently we will approximate the support of a single aggregation as $\Omega$. Using these assumptions we can rewrite Eq (10a) as a gradient flow of the form,

$$\frac{\partial g}{\partial t} = \nabla \cdot \left( g \nabla \left[ \frac{\delta E}{\delta g} \right] \right),$$

where

$$E[g] = \int_\Omega \frac{1}{2} g[Q_g \star g] + \frac{De^{-c}\gamma}{6} g^3 + De^{-c}(g \log(g) - g) \, dx, \tag{11}$$

where $E[g]$ represents an energy functional (can be thought of as a function of a function, see [40] for more details on gradient flows) with the minimisers satisfying

$$\frac{\delta E}{\delta g} = (Q_g \star g) + \frac{De^{-c}\gamma}{2} g^2 + De^{-c} \log(g) = \lambda.$$

Next, we follow the work of [31, 35, 41] and with a series of simplifying assumptions we

consider both the large and small mass limit in turn. First, we note that Eq (1) becomes

$$M = \int_\Omega \rho(x)\,dx = \int_\Omega g(x)\,dx.$$

To find the large mass limit, we begin with Eq (11) and assume that $g(x)$ is approximately rectangular and for a single aggregation that the support is far larger than the range of $e^{\frac{-|x|}{r}}$. This gives $e^{\frac{-|x|}{r}} \approx 2r\delta(x)$ (where $\delta(x)$ is the Dirac delta function), and therefore $Q_g \approx 2(R_g r_g - A_g)\delta(x)$. Using these assumptions we can estimate the maximum gregarious group density, $||g||_\infty$, as

$$||g||_\infty = \frac{3\left(-\left(R_g r_g - A_g\right) + \sqrt{\left(R_g r_g - A_g\right)^2 - \frac{4(De^{-c})^2\gamma}{3}}\right)}{2De^{-c}\gamma}, \tag{12}$$

with support,

$$||\Omega|| = \frac{2MDe^{-c}\gamma}{3\left(-\left(R_g r_g - A_g\right) + \sqrt{\left(R_g r_g - A_g\right)^2 - \frac{4(De^{-c})^2\gamma}{3}}\right)}. \tag{13}$$

The accuracy of this approximation is illustrated by Fig 1. We observe that within our model as $c$ increases so too does the maximum density of our locust formation. However, as the mass

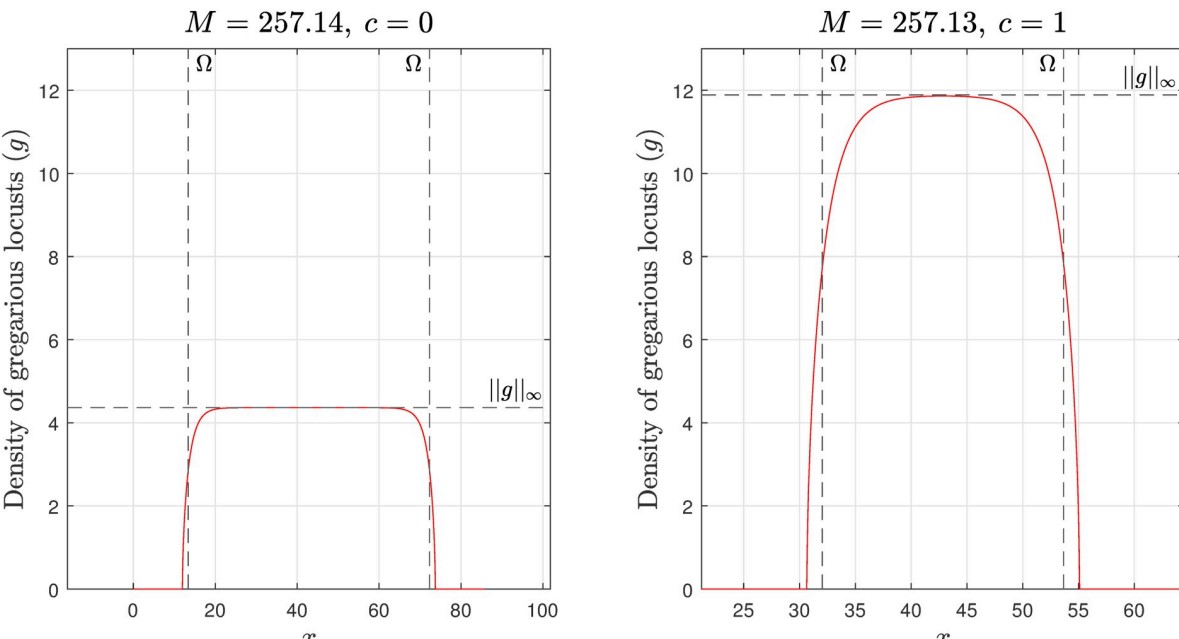

## Large mass limit

**Fig 1. Large mass limit with estimates for the max value and support.** The estimates of the max value and support are labelled $||g||_\infty$ and $\Omega$ respectively, with simulation results given by the red lines. For both the simulation and calculations $D = 0.01$, $\gamma = 60$, $R_g = 0.25$, $r_g = 0.5$, $A_g = 1$, and $c = 0$ and $1$. As the mass $M$ is increased the gregarious locust shape $g$ becomes increasingly rectangular as the maximum locust density does not depend on the total mass. In addition as the amount of food is increased from $c = 0$ on the left to $c = 1$ on the right, the maximum density for the gregarious locusts increases.

of locusts, $M$, increases the maximum density remains constant and the support $||\Omega||$ becomes larger. Finally, by using these derived relationships with field measurements of maximum locust densities we can estimate values of $\gamma$.

For the small mass limit, we begin with Eq (11) and approximate the social interaction potential using a Taylor expansion, $e^{\frac{-|x|}{r}} \approx 1 - \frac{|x|}{r}$ (giving $Q_g \approx (R_g - A_g) - |x|\left(\frac{R_g}{r_g} - A_g\right)$). In addition, to be able to solve the resulting equations we ignore the effect of linear diffusion within $\Omega$. While this gives a less accurate approximation it still shows the effect of food on maximum density. Under these assumptions, Eq (11) yields

$$E[g] = \int_\Omega \frac{1}{2} g\left(\left[(R_g - A_g) - |x|\left(\frac{R_g}{r_g} - A_g\right)\right] \star g\right) + \frac{De^{-c}\gamma}{6}g^3\, dx. \tag{14}$$

Following [35], we find the estimate of the maximum gregarious locust density, $||g||_\infty$, as

$$||g||_\infty = \sqrt[3]{\frac{3M^2\left(A_g - \frac{R_g}{r_g}\right)}{4De^{-c}\gamma}}, \tag{15}$$

with support,

$$||\Omega|| = B\left(\frac{2}{3}, \frac{1}{2}\right) \sqrt[3]{\frac{MDe^{-c}\gamma}{6\left(A_g - \frac{R_g}{r_g}\right)}}. \tag{16}$$

where $B$ is the $\beta$-function (for definition see [42], page 207).

The results of these approximations can be seen in Fig 2. While less accurate than those of the large mass limit, they illustrate that as the amount of food increases, so too does the maximum locust density. However, the effect is less pronounced than in the large mass case. It also demonstrates how the maximum locust density and support both increase with an increase in locust mass.

The accuracy of both the small and large mass approximations and the transition between the two can be seen in Fig 3 for both the maximum group density and support. In the simulations, we estimate the finite support, $\Omega$, as the region where $g > 0.01$. It is worth noting that the results for large and small mass limits likely apply to locust hopper bands and not just gregarious groups [23, 33].

**Linear stability analysis of homogeneous steady states.** In order to gain insights into the conditions under which groups can form, we investigate the stability of spatially-homogeneous steady states. In this analysis we perturb the homogeneous steady states by adding a small amount of noise. We then find under what conditions the small perturbations grow and are likely to lead to gregarious aggregations. As the calculation is somewhat lengthy, though standard, we omit the details here. They can be found in S2 Appendix.

We begin by defining the homogeneous steady states of $s$, $g$, and $c$, as $\bar{s}$, $\bar{g}$, and $\bar{c}$, with the total density given as $\bar{\rho} = \bar{s} + \bar{g}$. We again assume that $c$ does not deplete (i.e. $\kappa = 0$). As we are assuming either an infinite or periodic domain, we must redefine the global gregarious mass fraction, Eq (2), as

$$\phi_g(t) = \frac{g(t)}{\rho(t)}. \tag{17}$$

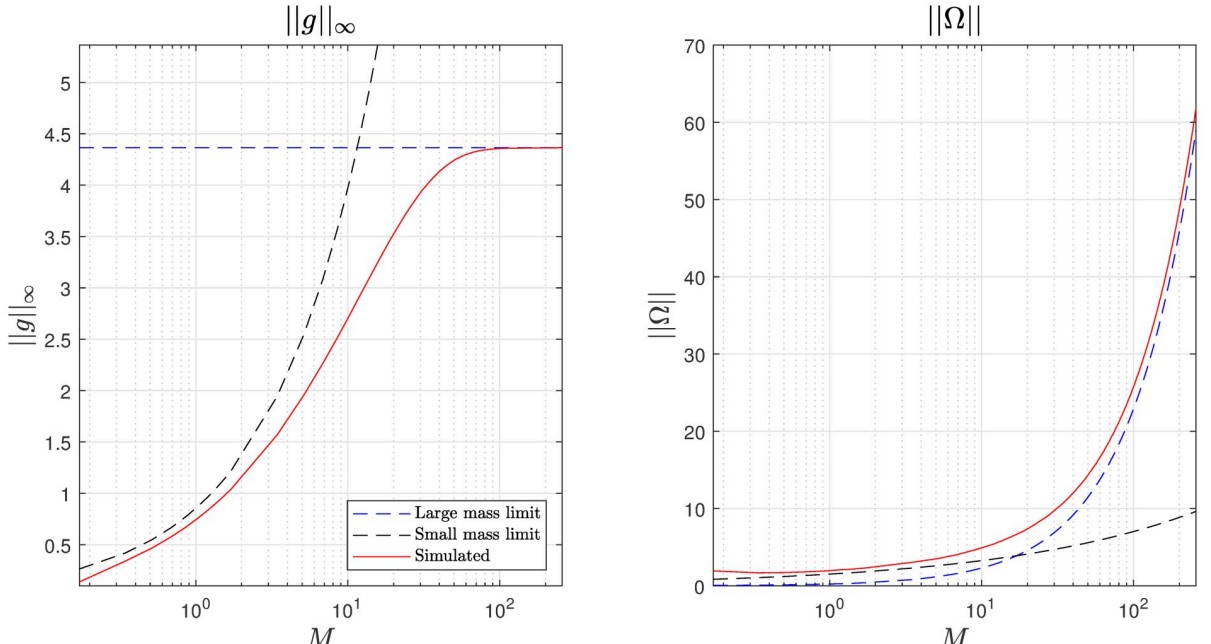

**Fig 2. Small mass limit with estimates for the max value and support.** The estimates of the max value and support are labelled $||g||_\infty$ and $\Omega$ respectively, with simulation results given by the red lines. For both the simulation and calculations $D = 0.01$, $\gamma = 60$, $R_g = 0.25$, $r_g = 0.5$, $A_g = 1$, and $c = 0$ and 1.

**Fig 3. Small and large mass limit estimates and simulated results for both the maximum group density (left) and support (right).** In the simulations we estimate the finite support, $\Omega$, as the region where $g > 0.01$.

By rewriting $\bar{s}$ and $\bar{g}$ in terms of this global gregarious mass fraction and the total density as

$$\bar{g} = \phi_g \bar{\rho}, \text{ and } \bar{s} = (1 - \phi_g)\bar{\rho},$$

we find the condition for group formation as

$$\phi_g > \bar{\phi}_g = \frac{\dfrac{De^{-\bar{c}}}{\bar{\rho}} + De^{-\bar{c}}\bar{\rho}\gamma + \hat{Q}_s}{\hat{Q}_s - \hat{Q}_g}, \tag{18}$$

where $\hat{Q}_s$ and $\hat{Q}_g$ are the Fourier transform of $Q_s$ and $Q_g$ respectively. From this, it can be seen that as $\bar{\rho}$ increases the gregarious fraction required for group formation increases. This effect is diminished as the amount of available food increases.

For our specific functions $Q_g = R_g e^{-\frac{|x|}{r_g}} - A_g e^{-|x|}$ and $Q_s = R_s e^{-\frac{|x|}{r_s}}$, taking the one dimensional Fourier transforms of $Q_s$ and $Q_g$ using the following definition,

$$\hat{f}(k) = \int_{\mathbb{R}^n} f(\boldsymbol{x}) e^{-ik\cdot\boldsymbol{x}} \, d\boldsymbol{x},$$

gives the following relationship,

$$\phi_g > \bar{\phi}_g = \frac{\dfrac{De^{-\bar{c}}}{\bar{\rho}} + De^{-\bar{c}}\bar{\rho}\gamma + 2R_s r_s}{2A_g - 2R_g r_g + 2R_s r_s}. \tag{19}$$

Interestingly, Eq (18) suggests there is also an upper limit on locust density for group formation. This would likely correspond with an environment so thick with locusts that there is insufficient room for aggregations to form. We can find this density by taking Eq (19) and substituting $\bar{\phi}_g = 1$ and solving for $\bar{\rho}$ as,

$$\bar{\rho} = \frac{(A_g - R_g r_g) + \sqrt{(A_g - R_g r_g)^2 - (De^{-\bar{c}})^2 \gamma}}{De^{-\bar{c}}\gamma} \approx \frac{2}{3}\|g\|_\infty,$$

where $\|g\|_\infty$ is maximum density for the large mass limit given in Eq (12).

Finally, we calculate if it is possible for a particular homogeneous density of locusts to become unstable (and thus form a gregarious aggregation). By calculating the homogeneous steady state gregarious mass fraction as,

$$\phi_g = \frac{f_2(\bar{\rho})}{f_1(\bar{\rho}) + f_2(\bar{\rho})},$$

then by combining with (19) we obtain an implicit condition for group formation as

$$\frac{f_2(\bar{\rho})}{f_1(\bar{\rho}) + f_2(\bar{\rho})} > \frac{\dfrac{De^{-\bar{c}}}{\bar{\rho}} + De^{-\bar{c}}\bar{\rho}\gamma + 2R_s r_s}{2A_g - 2R_g r_g + 2R_s r_s}. \tag{20}$$

In Eq (20), if the values on the left are not greater than those on the right then it is not possible for a great enough fraction of locusts to become gregarious and for instabilities to occur. As the value of the right hand side decreases as the amount of food increases, we can deduce that the presence of food lowers the required density for group formation.

**Time until group formation with homogeneous locust densities.** We also estimate time until group formation with homogeneous locust densities and a constant $c$. By assuming that $s$

and $g$ are homogeneous we can ignore the spatial components of Eq (10a) and (10b). We again denote the combined homogeneous locust density as $\bar{\rho}$ however now $\bar{\rho} = s(t) + g(t)$. Finally, assuming that $g(0) = 0$, we find the homogeneous density of gregarious locusts as a function of time is given by

$$g(t) = \frac{\bar{\rho} f_2(\bar{\rho})}{f_1(\bar{\rho}) + f_2(\bar{\rho})} \left( 1 - e^{-[f_1(\bar{\rho}) + f_2(\bar{\rho})]t} \right),$$

which we then solve for $t^*$ such that $g(t^*) = \bar{\phi}_g \bar{\rho}$, where $\bar{\phi}_g$ is given by Eq (18). This gives an estimation for time of group formation (i.e. the time required for the homogeneous densities to become unstable) as,

$$t^* = \frac{-\ln\left( 1 - \dfrac{\bar{\phi}_g(f_1(\bar{\rho}) + f_2(\bar{\rho}))}{f_2(\bar{\rho})} \right)}{f_1(\bar{\rho}) + f_2(\bar{\rho})}. \tag{21}$$

Thus, as increasing food decreases the gregarious mass fraction, $\bar{\phi}_g$, required for group formation it follows that it also decreases the time required for group formation.

**Conservation properties.** Another aspect of the model we investigate is what properties of locust densities the model conserves. By construction our model preserves the mass of locusts, i.e. Eq (1) is constant in time. In addition, using a similar method to [31] we show in S2 Appendix that in $\mathbb{R}^n$ and with a constant food source, i.e. $c(\boldsymbol{x}, t)$ is constant in space and time, the center of mass is also preserved. From this we can conclude that prior to group formation the locust center of mass is only moved due to non-uniformities in the food source.

## Numerical results

We now investigate both the effect of food on locust group formation and the effect of gregarisation on locust foraging efficiency in one dimension. In order to simulate our equations, we used a first order upwinding Finite Volume Scheme for the advection component with Fourier transforms to solve the convolution and central differencing schemes for the diffusion terms. We used an adaptive Runge-Kutta scheme for time. A full detailed derivation can be found in S3 Appendix.

**Parameter selection and initial conditions.** The bulk of the parameters, $R_s$, $r_s$, $R_g$, $r_g$, $A_g$, $k$, and $\delta$, have been adapted from [34] to our non-dimensionalised system of equations. We explore two parameter sets that we will term symmetric and asymmetric based on the time frame of gregarisation vs solitarisation. In the symmetric parameter set ($\delta = 1$, $k = 0.681$), gregarisation and solitarisation take the same amount of time and the density of locusts for half the maximal transition rate is lower for solitarisation. This is the default parameter set from Topaz et. al. [34] with an adjusted $k_1$ term that is calculated using Eq (20) and the upper range for the onset of collective behaviour as $\approx 65$ locusts/$m^2$ [34, 43]. This behaviour is characteristic of the Desert locust (*S gregaria*) [10].

In the asymmetric parameter set ($\delta = 1.778$, $k = 0.1$), solitarisation takes an order of magnitude longer than gregarisation, and the density of locusts for half the maximal transition rate is lower for solitarisation. This is the alternative set from Topaz et. al. [34]. The Australia plague locust (*Chortoicetes terminifera*) potentially follows this behaviour taking as little as 6 hours to gregarise but up to 72 hours to solitarise [44, 45]. The complete selection of parameters can be seen in Table 1.

At the densities we are investigating we will assume that the majority of movement will be due to locust-locust interactions rather than random motion, so we set our dimensional linear

**Table 1. Dimensionless parameters used in numerical simulations for both symmetric and asymmetric gregarisation-solitarisation.**

| Variable | Description | Symmetric Value | Asymmetric Value | Source |
|---|---|---|---|---|
| $k$ | Ratio of density of maximal phase transition rates | 0.681 | 0.1 | Eq (20) [24] [34] [10] [45] |
| $\delta$ | Ratio of maximal phase transition rates | 1 | 1.778 | [34] [10] [45] |
| $D$ | Linear diffusion coefficient | 2.041 | 2.041 | |
| $\gamma$ | Non-linear diffusion coefficient | 431.87 | 294.44 | Eq (12) [17] |
| $R_s$ | Strength of non-local solitarious repulsion | 1063.5 | 878.1 | [34] |
| $r_s$ | Range of non-local solitarious repulsion | 1 | 1 | [34] [24] |
| $R_g$ | Strength of non-local gregarious repulsion | 940.5 | 775.6 | [34] |
| $r_g$ | Range of non-local gregarious repulsion | 0.2857 | 0.2857 | [34] [24] |
| $A_g$ | Strength of non-local gregarious attraction | 2008.7 | 1658.6 | [34] |
| $\kappa$ | Food consumption rate | 0.09 | 0.18 | Eq (10c) |

diffusion term to be of the order $10^{-2}$, giving our non-dimensional linear diffusion as $D = 2.041$ for both symmetric and asymmetric parametrisation. Next, we estimate the maximum locust density as $\approx 1000$ locusts/$m^2$ [17] and adapt this to our one dimensional simulation as $||g||_\infty \approx 10\sqrt{10}$ locusts/$m$. Then using Eq (12) we find $\gamma = 431.87$ for the symmetric parameters and $\gamma = 294.44$ for the asymmetric parameters.

To estimate $\kappa$ we begin with Eq (10c) and set the nondimensionalised density of locusts to 1 ($\rho = 1$) (and $\rho = 0.5$ for the asymmetric parameters), we then want the locusts to consume approximately 70% of the food over the course of the simulation (i.e., $c$ transitions from $c = 1$ to $c = 0.30$). Solving for $\kappa$ we find $\kappa \approx 0.09$ (and $\kappa \approx 0.18$ for the asymmetric parameters).

Our spatial domain is the interval $x = [0, L]$, where $L = 3/0.14$ (this comes from non dimensionalising the domain used by [34]), with periodic boundary conditions (i.e., $s(0, t) = s(L, t)$). Our time interval is 12.5 units of time (in dimensional terms this is a 3m domain for a simulated 50 hours).

The initial locusts densities are given by

$$s(x, 0) = \frac{\rho_{\text{amb}}}{16.6}(16.6 + \mu) \text{ and } g(x, 0) = 0, \tag{22}$$

where $\rho_{\text{amb}}$ is a ambient locust density and $\mu$ is some normally distributed noise, $\mu \sim \mathcal{N}(0, 1)$. To ensure that simulations were comparable, we set-up three locust initial condition and rescaled them for each given ambient locust density. Finally, the initial condition for food is given by a smoothed step function of the form,

$$c(x, 0) = \frac{F_M}{2\zeta}\left[\tanh\left(\alpha\left[x - \left(x_0 - \frac{\zeta}{2}\right)\right]\right) - \tanh\left(\alpha\left[x - \left(x_0 + \frac{\zeta}{2}\right)\right]\right)\right], \tag{23}$$

with $\alpha = 7$, $x_0 = L/2$, $F_M$ being the food mass and $\zeta$ being the initial food footprint. We will also introduce $\omega = 100\zeta/L$ which expresses the size of the food footprint as a percentage of the domain.

**The effect of food on group formation.** To investigate the effect that food had on locust group formation, we ran a series of numerical simulations in which the total number of locusts and the size of food footprint were varied, while the total mass of food remained constant. The food footprint ranges from covering 2.5% of the domain to 50% of the domain ($\omega = 2.5\%$ to $\omega = 50\%$). For the symmetric parameters four food masses were tested, $F_M = 1.5, 2, 2.5$ and 3, and for the asymmetric variables two food masses were tested, $F_M = 1.5$ and 3. As a control we also performed simulations with both no food present and a homogeneous food source, represented by $\omega = 0\%$ and $\omega = 100\%$ respectively, for each ambient locust density.

We varied the ambient locust density ranging from $\rho_{amb} = 0.8$ to $\rho_{amb} = 1.4$ for the symmetric parameters. This range was selected based on Eq (20) so that in the absence of food group formation would not occur. In each simulation, the solitarious and gregarious populations very quickly tend to an almost smooth and symmetric distribution around the food, however a small quantity of noise persists across the population and this breaks the symmetry leading to group formation. As we found in certain cases the initial noise had an effect on whether a group would form we ran three simulations for each combination of $\rho_{amb}$, $\omega$, and $F_M$ with varied initial noise and took the maximum peak density across the three simulations.

For the asymmetric variables we varied $\rho_{amb}$ from $\rho_{amb} = 0.3$ to $\rho_{amb} = 0.55$, to test the effect food had on the time frame of group formation. From Eq (20) in the absence of food there should be group formation in the upper half of this density range. However Eq (21), suggests this will only occur outside or right at the end of our simulated time frame. We ran a single simulations for each combination of $\rho_{amb}$, $\omega$, and $F_M$.

The results for the symmetric parameter experiments are displayed in Fig 4. The plots show the peak gregarious density of the three simulations for each of the varying food footprint sizes

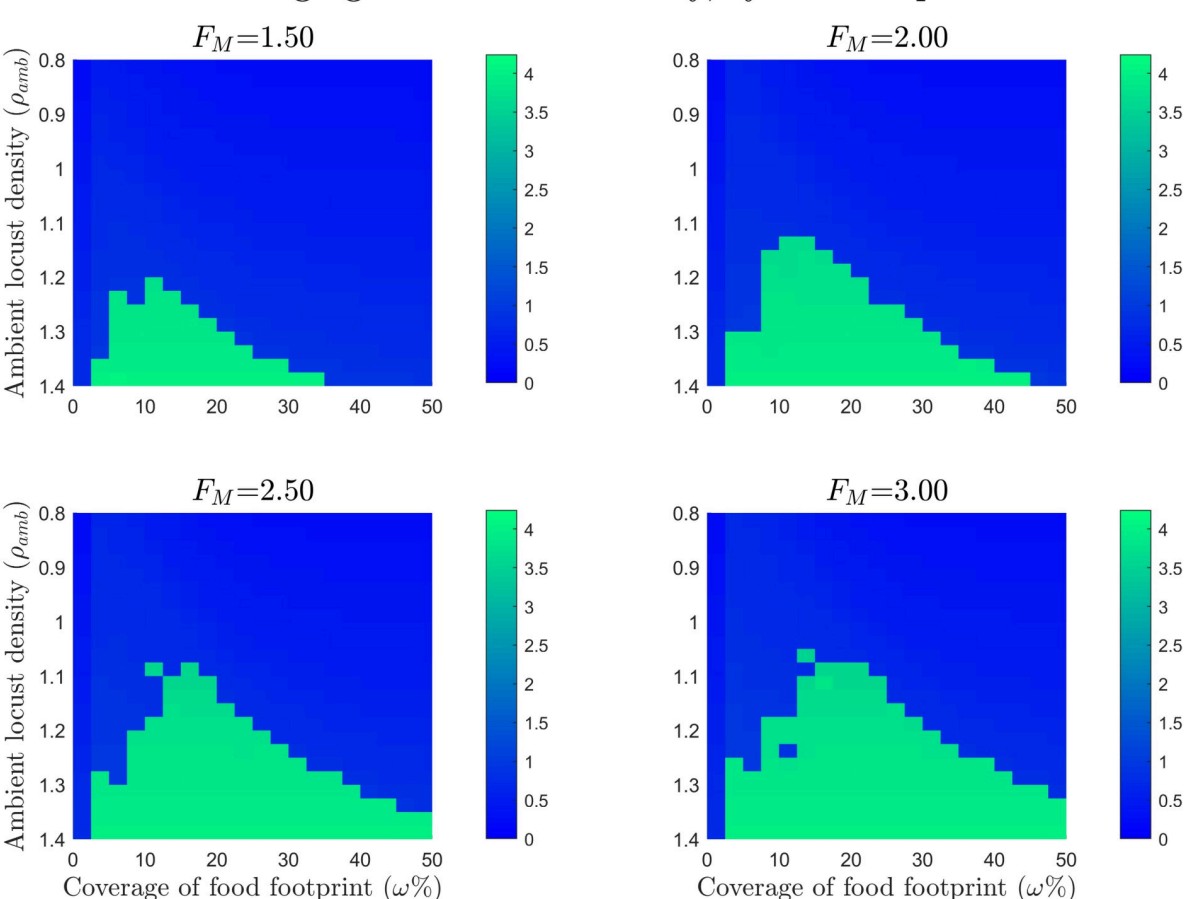

**Fig 4. Maximum gregarious locust density for the symmetric gregarisation parameters with varying food footprint sizes and initial ambient locust densities.** For the simulations, $x = [0, 3/0.14]$ with periodic boundary conditions and $t = [0, 12.5]$. The initial condition for locust densities is given by Eq (22) and food initial conditions are given by Eq (23). Ambient locust density ranges from $\rho_{amb} = 0.8$ to $\rho_{amb} = 1.4$, food footprint ranges from $\omega = 0\%$ to $\omega = 50\%$, the food mass $F_M = 1.5, 2, 2.5$ and 3, and the consumption rate $\kappa = 0.09$. The plots show the maximum peak gregarious density for the varying food footprint sizes and ambient locust densities, in the blue regions there was no group formation and in the green regions there was successful group formation. From this we can deduce that food lowers the required locust density for group formation and this is more pronounced within an optimal food width.

## Maximum gregarious locust density, asymmetric parameters

**Fig 5. Maximum gregarious locust density for the asymmetric gregarisation parameters with varying food footprint sizes and initial ambient locust densities.** For the simulations, $x = [0, 3/0.14]$ with periodic boundary conditions and $t = [0, 12.5]$. The initial condition for locust densities is given by Eq (22) and food initial conditions are given by Eq (23). Ambient locust density ranges from $\rho_{amb} = 0.3$ to $\rho_{amb} = 0.55$, food footprint ranges from $\omega = 0\%$ to $\omega = 50\%$, the food mass $F_M = 1.5$ and 3, and the consumption rate $\kappa = 0.18$. The plots show the maximum peak gregarious density for the varying food footprint sizes and ambient locust densities. In the blue regions there was no group formation and in the green regions there was successful group formation. From this we can deduce that food lowers the required time forgroup formation and again this is more pronounced within an optimal food width.

and ambient locust densities. In the blue regions there was no group formation, whilst in the green regions indicate successful group formation. It can be seen in the plots that as the food mass is increased the minimum required locust density for group formation decreases. This effect is more pronounced within an optimal food width and this optimal width increases as the amount of food increases.

The results for the asymmetric parameter experiments are displayed in Fig 5. Again, green indicates successful group formation and blue indicates no group formation. It can be seen in these plots that with no food present a group failed to form within the simulated time. From this we can infer that food also decreases the required time for group formation, again there is an optimal food width for this effect.

We can delve deeper into the results by looking at a representative sample of simulations in Fig 6. In these simulations $\rho_{amb} = 1.2$, $\kappa = 0.09$, and $F_M = 1.5$, with food footprints $\omega = 7.5\%$, 10%, and 12.5% as well as with no food present. In the simulations in which food is present, prior to group formation gregarious locusts aggregate at the center of the food. If the food source is too narrow ($\omega = 7.5\%$, $t = 3$) there is an attempt at group formation but the gregarious mass is too small and the food source has not been sufficiently depleted so a large portion remains within the food source, thus the group does not persist. If the food is too wide ($\omega = 12.5\%$) the gregarious locusts simply cluster in the center of the food and do not attempt group formation. However, if the food width is optimal ($\omega = 10\%$) there is a successful group formed, this is seen as clump or aggregation of gregarious locusts in the final plot.

**The effect of gregarisation on foraging efficiency.** It is also possible to investigate the effect of gregarisation on foraging efficiency. Using [46–48] as a guide we first calculate the per capita contact with food for solitarious and gregarious locusts, respectively as

$$\eta_s(t) = \frac{1}{M} \int_0^L \frac{c(x,t)s(x,t)}{(1 - \phi_g(t))} \, dx \text{ and } \eta_g(t) = \frac{1}{M} \int_0^L \frac{c(x,t)g(x,t)}{\phi_g(t)} \, dx,$$

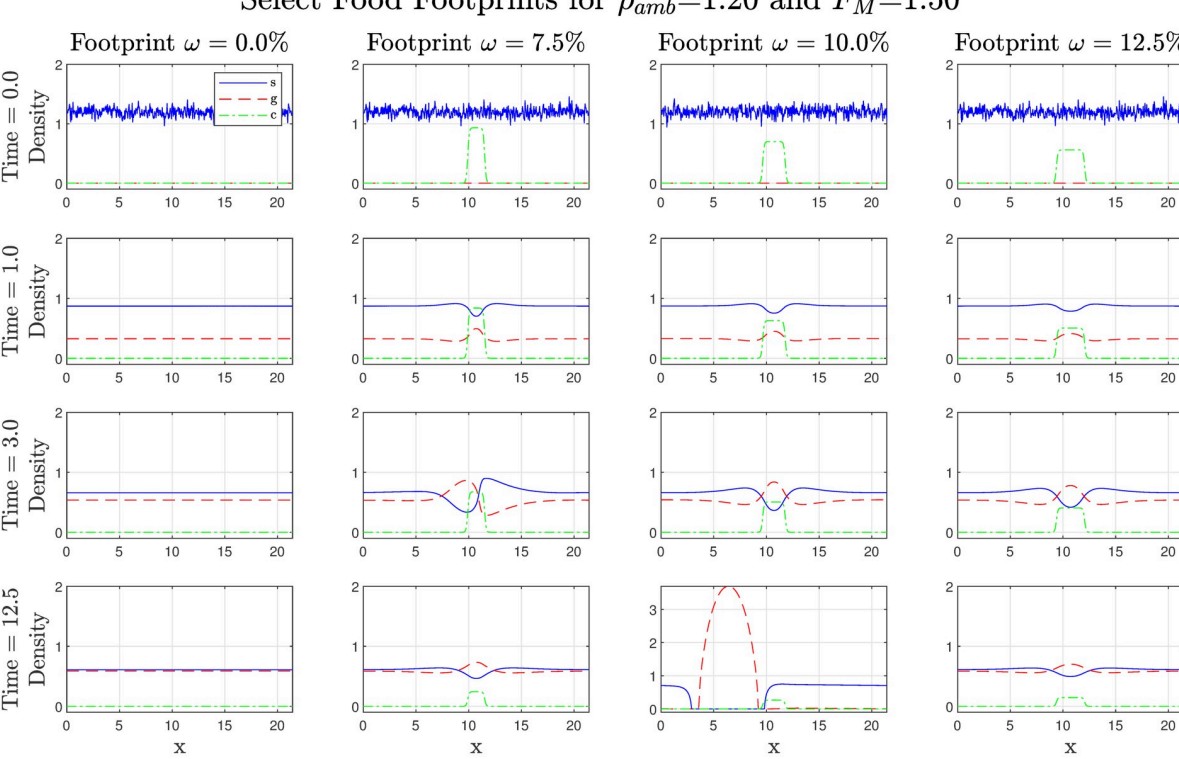

**Fig 6. A selection of plots showing the effect of food distribution on gregarisation and locust group formation with symmetric parameters.** In these simulations $\rho_{amb} = 1.2$, $\kappa = 0.09$, and $F_M = 1.5$ with $\omega$ = 7.5%, 10%, and 12.5%, as well as with no food present (labelled $\omega$ = 0%). In the plots, blue is solitarious, red is gregarious, and green is food. If the food source is too narrow ($\omega$ = 7.5%, $t$ = 3) there is an attempt at group formation but the gregarious mass is too small and a large portion remains within the food source, thus the group does not persist. If the food is too wide ($\omega$ = 12.5%) the gregarious locusts simply cluster in the center of the food and do not attempt group formation. Finally, if the food width is optimal ($\omega$ = 10%) there is a successful group formed, this is seen as clump or aggregation of gregarious locusts in the final plot.

where $M$ is given by (1). We then calculate the instantaneous relative advantage at time $t$ as

$$b(t) = \frac{\eta_g(t)}{\eta_s(t)}. \tag{24}$$

For full reasoning behind the validity of these metrics of foraging efficiency see S4 Appendix. We then select a range of food footprints, $\omega$%, and two food masses, $F_M$, for a fixed ambient density of locusts, $\rho_{amb} = 0.95$, from the previous simulations. We record the gregarious mass fraction and instantaneous relative advantage as functions of time and plot these against each other in Fig 7. By looking at the instantaneous relative advantage versus the global gregarious mass fraction prior to group formation in Fig 7, it can be seen that as the gregarious mass fraction increases so too does the foraging advantage of being gregarious. Thus, as a greater proportion of locusts become gregarised it is more advantageous to be gregarious. This effect is increased by the mass of food present but is diminished by the size of the food footprint to the point where no advantage is conferred when the food source is homogeneous. This effect is visualised in Fig 6, as prior to group formation gregarious locusts aggregate in the center of the food mass and displace their solitarious counterparts.

## Instantaneous relative advantage vs gregarious mass fraction

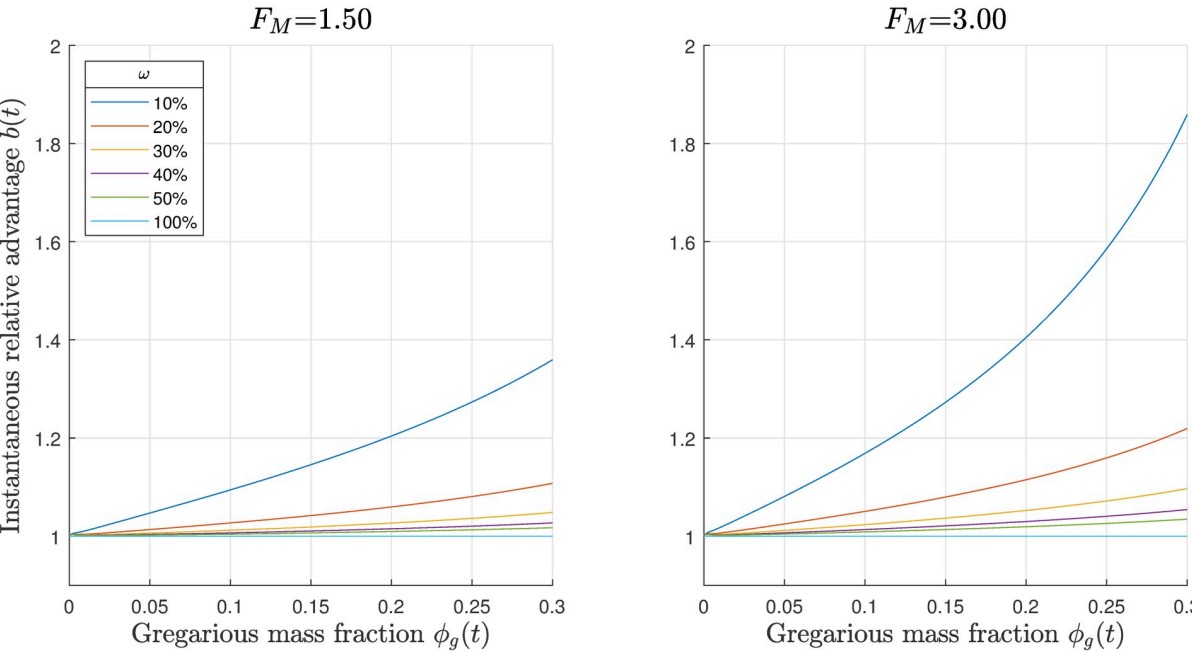

**Fig 7. Instantaneous relative advantage of gregarious locusts vs gregarious mass fraction at various food footprints and food masses.** In these simulations $\rho_{amb} = 0.95$ and $\kappa = 0.09$, with the symmetric parameter set. The homogeneous food source is labelled $\omega = 100\%$. It can be seen that as the gregarious mass fraction increases so too does the foraging advantage of being gregarious, this effect is increased by the mass of food present but is diminished by the size of the food footprint.

## Discussion

Locusts continue to be a global threat to agriculture and food security, and so insights into the hopper band formation process that can help predict and control outbreaks is of great importance. In this paper we presented a continuum model that includes non-local and local inter-individual interactions and interactions with food resources. This model extends the model of Topaz et. al. 2012 [34] for locust gregarisation to include food interactions and local repulsion. By analysing and simulating our new model we have found that food acts to: increase maximum locust density, lower the gregarious fraction required for group formation (an important precursor to locust hopper bands), and decreases both the required density and time for group formation with this effect being more pronounced at some optimal food width.

Analytical investigations of our model shows that a spatially uniform and temporally constant food source has a variety of effects has on locust behaviour. Firstly, by considering a purely gregarious population we found that the maximum locust density is affected by the amount of food present, in that increasing food leads to increased maximum density. Then, by performing a linear stability analysis we found the gregarious mass fraction required for group formation depends on both the ambient locust density and the amount of food present, with increasing food decreasing the required gregarious mass fraction. Using this relationship we then found that the presence of food lowers both the required time and density of locusts for group formation, and interestingly that our model also has a theoretical maximum locust density for group formation. Finally, we have also shown that the center of mass of locusts is not dependent on the locust-locust interactions we explored, so prior to non captured interactions such as alignment the movement of the center of mass is driven by food. In simulations this

was seen when prior to group formation gregarious locusts aggregated at the center of the food source.

Then using numerical simulation techniques we confirmed in our model that similar to previous studies highly clumped food sources lead to a greater likelihood of gregarisation [20]. However, we found that there may exist an optimal width for these food clumps for group formation. Similar to our analytic investigations, food was shown to lower the required density for group formation via the symmetric parameters and the required time via the asymmetric parameters. We also found that the optimal width is dependent on the amount of food present relative to the locust population. This effect appears to be brought about by the depletion of the food source, if the food source is not sufficiently depleted, then a gregarious group will fail to form because a portion of the gregarious population will remain on the food. In addition, by looking at the relative foraging advantage of gregarious locusts in our simulations we found that as the gregarious mass fraction increases so too does the foraging advantage of being gregarious. This effect is increased by the mass of food present but is diminished by the size of the food footprint to the point where no advantage is offered with a homogeneous food source.

In 1957 Ellis and Ashall [49] found that dense but patchy vegetation promoted the aggregation of hoppers and that sparse uniform plant cover promoted their dispersal. While there are various explanations about the costs and benefits of group living [50], it is less well understood for phase polyphenism. In addition to studies having shown benefits in terms of predator percolation [51] or in relation to cannibalism [52]. Our study, in line with recent studies about solitary and social foraging in complex environments [53] and Ellis and Ashall observations [49], provide another possible avenue of exploration for the advantage of phase polyphenism.

As with many models, ours required a variety of simplifying assumptions to keep the mathematics tractable, which limits the direct biological relevance of the model at present. While our model is most applicable in the stage prior to hopper band formation and does not properly capture the movement of hopper bands, these results presented can give guidance on how higher order models might behave [23, 33]. With this in mind, there are many ways that the model could be further developed. First, by having locust behaviours dependent on time, levels of hunger [54], and/or the inclusion of a heterogeneous age structure. Differing local locust-locust and locust-food interactions between solitarious and gregarious populations. Finally, using a higher order model that is able to capture collective movement mechanisms such as alignment or pursuit/escape interactions [55].

Finally, preventative methods are the key to improving locust control. This includes the ability to predict mass gregarisation according to resource distribution patterns so that the area searched for locusts is reduced and control efforts are deployed in high risk areas early on [18]. Further exploration of our results has the potential to improve predictive gregarisation models and early detection efforts by further increasing our understanding of the link between gregarisation and vegetation (resource) distribution (the latter becoming increasingly easy to quantify during field surveys, and aerial surveys including drones and satellite imagery [21, 45]). Future research could focus on developing decision support systems integrating predictive gregarisation models and GIS data from surveys.

## Supporting information

**S1 Appendix. Detailed derivation of local flux.** The full detailed derivation of the flux terms based on local interactions given in the model derivation section.
(PDF)

**S2 Appendix. Detailed analytic results.** The full detailed derivations of the analytic results given in the PDE model analysis section.
(PDF)

**S3 Appendix. Numerical scheme.** The full detailed derivation of the numerical scheme used for simulating the numerical results.
(PDF)

**S4 Appendix. Foraging efficiency.** Derivation of instantaneous relative advantage from the marginal value theorem.
(PDF)

## Author Contributions

**Conceptualization:** Fillipe Georgiou, Camille Buhl, J. E. F. Green, Bishnu Lamichhane, Ngamta Thamwattana.

**Formal analysis:** Fillipe Georgiou, Camille Buhl, J. E. F. Green, Bishnu Lamichhane, Ngamta Thamwattana.

**Methodology:** Fillipe Georgiou, Camille Buhl, J. E. F. Green, Bishnu Lamichhane, Ngamta Thamwattana.

**Writing – original draft:** Fillipe Georgiou.

**Writing – review & editing:** Fillipe Georgiou, Camille Buhl, J. E. F. Green, Bishnu Lamichhane, Ngamta Thamwattana.

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
