## [Decision Letter · Decision Letter 0]

29 Dec 2020

Dear Mr Georgiou,

Thank you very much for submitting your manuscript "Modelling locust foraging: How and why food affects hopper band formation" for consideration at PLOS Computational Biology.

As with all papers reviewed by the journal, your manuscript was reviewed by members of the editorial board and by several independent reviewers. In light of the reviews (below this email), we would like to invite the resubmission of a significantly-revised version that takes into account the reviewers' comments.

As can be seen from their reports below, all 3 reviewers found this work interesting and the methodology robust. However some major concerns were also raised. In particular Reviewer 1 makes the valid point that the connection between the biology and the model is very loose: unlike in real locust swarms, the bands in the model do not intrinsically move (their centre of gravity does not move in the absence of non—uniformities in food, which is not the case in real swarms) and the model is limited to 1 dimension, limiting its applicability.

I agree with these points, but I—and all reviewers—value the mathematical approach employed which, although it does not relate well to real locusts, provides value in the importance of addressing the issues of scaling in such biological systems. That is, one could argue that the approach here has failed to replicate even the basic phenomena seen in real swarms, but this in itself serves the purpose to highlight how and why these major discrepancies may have arisen. Consequently I would strongly recommend the authors reconsider how the paper is structured and should use the locusts more of a source of inspiration, and they should discuss openly such major difference seen between the natural and model systems. This model should not be ‘sold’ as a model of actual locusts, but rather is inspired by them and acts as a valid and useful exercise in determining how simple rules scale to macroscopic features. This would, in my mind, resolve the issues raised both by reviewer 1 and my own feelings about the manuscript.

One further issue that needs to be addressed, and readily can, is the use of “evolution”. It should be removed. This work does not consider evolutionary dynamics in an appropriate way, and as noted above this isn't really a viable model of real swarms nor the selection pressures on them. The costs and benefits could be explored in the way presented here, but please note the valid concern of reviewer 1 who notes that you may simply be getting our what you put in. Note also that there has been work on the evolution of phase polyphenism and cannibalism (Guttal et al. (2012) “Cannibalism can drive the evolution of behavioural phase polyphenism in locusts” Ecology Letters) which could be considered when discussing costs and benefits, and previous work that has modelled evolution of locust swarming.

The points above are major, but they are, in my opinion, not too difficult to fix since they change the way the story is told as opposed to changing the actual results. But this will be important to do. In addition please make the figures more readable, and it would be helpful to have a table of terms and values used. Below I include the detailed comments from the reviewers which should also be responded to point by point.

We cannot make any decision about publication until we have seen the revised manuscript and your response to the reviewers' comments. Your revised manuscript is also likely to be sent to reviewers for further evaluation.

Sincerely,

Iain Couzin

Guest Editor

PLOS Computational Biology

Stefano Allesina

Deputy Editor

PLOS Computational Biology

As can be seen from their reports below, all 3 reviewers found this work interesting and the methodology robust. However some major concerns were also raised. In particular Reviewer 1 makes the valid point that the connection between the biology and the model is very loose: unlike in real locust swarms, the bands in the model do not intrinsically move (their centre of gravity does not move in the absence of non—uniformities in food, which is not the case in real swarms) and the model is limited to 1 dimension, limiting its applicability.

I agree with these points, but I—and all reviewers—value the mathematical approach employed which, although it does not relate well to real locusts, provides value in the importance of addressing the issues of scaling in such biological systems. That is, one could argue that the approach here has failed to replicate even the basic phenomena seen in real swarms, but this in itself serves the purpose to highlight how and why these major discrepancies may have arisen. Consequently I would strongly recommend the authors reconsider how the paper is structured and should use the locusts more of a source of inspiration, and they should discuss openly such major difference seen between the natural and model systems. This model should not be ‘sold’ as a model of actual locusts, but rather is inspired by them and acts as a valid and useful exercise in determining how simple rules scale to macroscopic features. This would, in my mind, resolve the issues raised both by reviewer 1 and my own feelings about the manuscript.

One further issue that needs to be addressed, and readily can, is the use of “evolution”. It should be removed. This work does not consider evolutionary dynamics in an appropriate way, and as noted above this isn't really a viable model of real swarms nor the selection pressures on them. The costs and benefits could be explored in the way presented here, but please note the valid concern of reviewer 1 who notes that you may simply be getting our what you put in. Note also that there has been work on the evolution of phase polyphenism and cannibalism (Guttal et al. (2012) “Cannibalism can drive the evolution of behavioural phase polyphenism in locusts” Ecology Letters) which could be considered when discussing costs and benefits, and previous work that has modelled evolution of locust swarming.

The points above are major, but they are, in my opinion, not too difficult to fix since they change the way the story is told as opposed to changing the actual results. But this will be important to do. In addition please make the figures more readable, and it would be helpful to have a table of terms and values used. Below I include the detailed comments from the reviewers which should also be responded to point by point.

I look forward to seeing a new version of the manuscript.

Iain Couzin, Guest Editor

Reviewer's Responses to Questions

**Comments to the Authors:**

Reviewer #1: The manuscript entitled “Modelling locust foraging: How and why food affects hopper band formation” by Georgiou et al describes analytical and numerical studies of a PDE model for coarse-grained locust swarm dynamics. The model describes the evolution of the density of locusts (either gregarious or solitarious) and food. Analytically, it is shown that, assuming homogeneous and constant densities, the model admits maximal and minimal mass limits. The stability of the homogeneous and constant solutions is also studied. Numerically, the authors solve a 1D version of the model. In particular, the dependence of results on the initial food distribution is studied.

Unfortunately, I cannot recommend publication of the manuscript to PLoS Comput Biol. The main reasons are as follows.

1. I find the paper lacks a well-defined focus or goal. This starts with the abstract, which reads like an introduction. The main results are not explained clearly. The figures are difficult to interpret (small font, the captions do not explain what the reader is supposed to look at). While the many parameters are defined in the text, the reader should not be expected to remember everything (for example, write the meaning of the parameters in Table 1). The discussion is somewhat scattered and lacks a concrete scientific claim. These comments may be addressed by a through revision.

2. While the mathematics seems of high quality and the biological background is solid and well explained, the connection between the two is loose. Here are a few examples.

i. There is no clear definition or explanation what the authors mean by band formation. As far as I understand, they refer to the case in which the fixed, homogeneous solution is unstable. Other studies, e.g. [17] and [33], refer to a moving band as a traveling wave solution with a given density profile. Therefore, the rigorous stability analysis, while mathematically valid, cannot be applied to band formation in realistic locust swarms.

ii. The numerical simulations are one dimensional. This is perfectly reasonable for academic interests or for modelling marching in circular arenas. However, the relation to realistic moving bands is marginal. In particular, I cannot see how 1D simulations can be used to infer the evolutionary advantages of the specific band width observed in nature.

iii. Fig. 6 shows situations in which the gregarious and solitarious locusts occupy the epoch of the band (at x=10) simultaneously. While I understand this result mathematically, I am not sure how realistic is. Depending on species, solitarious individuals may be present in dilute areas of a band, e.g. in the rear [17], but not at the most dense parts.

iv. The conclusions on the benefits of gregariousness for foraging seem to be a direct consequence of some modelling assumptions. In simulations, the initial food is concentrated in a given region (a single patch) and both gregarious and solitarious locusts are attracted to it. Therefore, if gregarious animals are attracted to each other while solitarious animals are repelled, then more gregarious individuals will be around the food. I expect this result will change if several food patches are introduced.

Minor comments:

1. Eqs. (5) and (6) are the same as (1) and (2).

2. Below (6): (3a) is referred to twice.

3. Several assumptions underlying the model should be explained and discussed. Making simplifying assumptions is fine. However, in the very least, it should be pointed out that they may not hold in realistic locust swarms. Here are several examles:

i. Page 8: I do not see any reason to assume the law of mass action except that it is simple. In particular, the system is not at equilibrium.

ii. Locust phase polyphenism is continuous.

iii. The model assumptions regarding the effect of food on the movement of locusts should be discussed in light of the following recent highly relevant paper: Dkhili, J., Maeno, K. O., Hassani, L. M. I., Ghaout, S., & Piou, C. (2019). Effects of starvation and vegetation distribution on locust collective motion. Journal of Insect Behavior, 32(3), 207-217.

iv. Page 9, last line: Relate the constants taken from [31] to the proper experiments.

v. Why is it OK to ignore the diffusion?

4. Page 12: Explain the meaning of small and large mass limits. It is not clear.

5. The sub-section showing that the center of mass does not move is a nice exercise, but the result is trivial considering the symmetry of the model and initial conditions. The two-page derivation can be put in an appendix.

6. Figs 4 and 5: I see only two colors – blue and green. Are there only two states or a continuum of values, as the color-bar suggests?

Reviewer #2: This manuscript derives and analyzes a mathematical model (PDE) for locust foraging and interaction that extends the work of Topaz et al. 2012. The model includes both gregarious and solitarious locust densities, a food resource density, and both local and non-local aggregation and repulsion effects. The paper’s focus is primarily on mathematical analysis of the model and is quite extensive while staying within the context of those qualitative features that are relevant to the biological system being represented. My feeling is that the manuscript is a fine contribution to the field and should be accepted following a number of mostly minor revisions. The main topics of my comments center around the derivation of the spatial flux terms in the model along with a pervasive need to clarify the language at various points in the paper.

1. Starting around line 121, there is some general confusion about this term T with the subscript i and the plus/minus superscript (here, I will just refer to it as T). On line 121 it is referred to as a rate. Then on line 124, alpha and beta are introduced and said to be probabilities of movement. This is really vague… movement of what, going where? And how does this result in a rate on the left hand side of the equation? What are the units of this rate? Then on line 131, tau is defined via a law of mass action, but there is no constant of proportionality. Perhaps the authors intend that it will be subsumed into beta, but then beta should not be a probability. Then on line 132, T is called a transition probability. So which is it? A transition probability, or a rate? The distinction matters: T is a nonlinear function of rho. So if we are being asked to consider rho as a stochastic variable (being a sum of the stochastic variables s and g), the expected value of the transition probability T is not equal to the RHS of the equation using the expected value of rho. For a nice explanation on this, see the seminal paper by Mollison (1977), “Spatial contact models for ecological and epidemic spread” in J. R. Statist. Soc. B, particularly the section Relations Between Stochastic and Deterministic Models. So I would ask that the authors please make this section more rigorous. I would also request that there be more detail in the derivation of the continuum limit - this is quite a jump since there is both a spatial and temporal limit occurring, and it is not at all clear how alpha and beta (which I’m already fuzzy on) translate to D and gamma.

2. Equation 14: I am really not sure where this E expression is coming from… it seems to appear a bit out of the blue. Can you provide some more guidance? Also, I would suggest writing E(g) instead of E[g], as with brackets this often means “the expectation of g”.

Comments relating to language:

- Abstract: do you want a colon after “These are” in the third line…?

- Line 59: “It is based on”

- Equation (5) and (6) and the language surrounding them are a repeat from Equation (1) and (2)

- Line 102: Bold J for non-local flux

- Line 105: grammar

- Line 114: grammar (comma splice)

- Line 134: if -> of

- Line 162: center of mass

- Line 164-166. The meaning of this sentence is not clear to an external reader. What does it mean to find these maximums for “large” and “small” numbers of locusts? This makes sense later on, but the first time you refer to it is here, and it’s really confusing.

- Line 167: What is the practical effect of assuming c is constant and not depleting? That is, what biological scenario does this correspond to and why can/should we assume this? I’m just not sure why this assumption should be made, and a sentence or two giving context would be helpful.

- Line 171: grammar (comma splice)

- Line 203-204: wording

- Line 208: Same comment as line 167

- Line 210: Odd wording given the previous sentence. Please make this paragraph more easy to follow.

- Line 215: What is an “upper locust density”? Be a bit more descriptive..?

- Line 221-222: This was a bit hard to follow in my reading, perhaps because it sounds a little like “when there are a lot of locusts in an area, there can’t be a lot of locusts in an area”. Maybe just provide some context. Biologically, why should we expect a maximum homogenous locust density under which locust aggregations can form? And make it clear what you mean by “locust aggregations.”

- Line 257: “we find an alternate expression”

- Line 268: comma splice

- Fig. 4 caption: why was the final time chosen to be 12.5? This seems like a rather odd number. Also, this wasn’t mentioned in the text anywhere… just in the caption?

- omega: if it’s a percent, write the numerical value with a percent sign. E.g. omega = 0% to 50%. This helps your reader follow what these things are. Same comment for all numerical expressions of omega down to Fig. 7.

Reviewer #3: Summary:

This paper makes a nice contribution to the locust modeling literature; it is the only agent-based model I know of at the moment that includes attraction/repulsion, resources and phase changes (gregarisation) and starts providing a framework for studying how food distribution can mediate the large-scale gregarisation that mediates the formation of hopper bands.

Perhaps one of the greatest challenges at present in the mathematical modeling of biological population models is developing models that can give robust insight biological processes; the difficulties of identifying appropriate models and parameters from the biology and then sampling the often high-dimensional parameter space should not be underestimated. As such, the fact that this paper leaves me asking for more is a sign of its strength. The conclusion here is that i) resource distribution can mediate the threshold for large-scale gregarisation and ii) there is a foraging advantage for gregarized locusts in certain resource distribution scenarios.

Below is a list of points/errata I would encourage the authors to address; none of them will change the results/narrative of the paper, but they will improve its readability

I believe that after the authors address these points the paper will be a strong contribution to PLOS Comp Bio.

Detailed comments:

• Throughout the paper – the authors should make the spelling of gregarize/gregarise and solitarize/solitarise (and gregarisation/solitarisation etc.) uniform.

• Abstract: The authors write “It is these short time-scale locust- resource relationships and their effect on hopper band formation that are of interest.” Of interest to whom? Passive voice here is confusing/uninformative.

• Line 16: “ . . . process of transition called gregarisation.” Could the authors add another sentence of explanation here – state clearly what causes gregarisation and perhaps address the timescale on which it occurs, note the process is reversible, and note that solitarisation may take place on a longer timescale. This will foreshadow some of the results investigated later.

• Line 48 -51: Do the authors wish to make clear in the introduction that there are many species of locusts and that the observations referred to here are specifically for S. gregaria?

• Line 67-68: There is a potential for confusion here of the meaning of multiple species (mathematical vs. biological) – perhaps multiple component/multiple populations might be better. In the abstract the authors use the language “multiple populations” – why not use that language here also.

• Line 100: For the global mass fraction to make sense you’ve assumed the total locust mass is finite. Is the domain finite here? Periodic? As you use this quantity later in the stability calculation for an infinite domain of constant mass fraction it might be worth a comment to make the definition consistent.

• Line 102: Another assumption in this model is that the behaviors are constant in time. If we are considering action over multiple days, that certainly isn’t true (for example, locusts certainly feed more and are more active in the day when it is warmer). To be clear, I think from a modelling perspective this is fine, but I believe it should be called out as an assumption.

• Lines 110-112 appear to be a repeat of lines 99-101.

• Lines 113 and 114: It says “in Eq (3a) 113 and Eq (3a)” but probably means “in Eq (3a) 113 and Eq (3b)”

• Line 132: The authors assume that the local behaviors are the same for gregarious and solitarious locusts. While one could argue the diffusive type behavior (which is not social) might be the same (even though gregarious locusts I believe are more active) the second term (which is related to collisions) arguably should be different for the two behavioral phases. I’m all in favor of simplifying assumptions, but the authors might address this point.

• Equations 8a & 8b: I am reasonably sure that there is an extra divergence operator in this expression – shouldn’t they be vectors (and therefore multiples of the gradients in the spatial variable)?

• Lines 134-141: I’m a little confused here – the authors have in essence included both local and non-local repulsion. Repulsion is sometimes thought of biologically as collision avoidance (which is the essence of the local derivation here). Perhaps the authors could clarify why they include both effects?

• Line 246-247: I suspect the statement about boundary conditions and center of mass here is incorrect – consider just the diffusion equation in 1D. You need <x,\\rho_xx> to vanish – integrating by parts twice yields boundary terms like x\\rho_x - \\rho suggesting the correct statement is that you need both \\rho and \\rho_x to vanish at the boundary of the domain. Moreover, if the diffusion constant (D) here is non-zero the support of the solution nearly certainly becomes the entire domain after an infinitesimal amount of time. So the statement here is probably only correct if: 1) D=0 and the solution is compactly supported or 2) D>0, the domain is infinite and the mass is finite (an analyst might want a bound on the density gradient also but an applied mathematician should be happy).

• Line 284: Returning to the question of the two repulsive behaviors it might be interesting to investigate the relative sizes with these parameters of the effects of diffusion, collision avoidance (local non-linear repulsion) and non-local repulsion in the stability criteria below (225) and in numerical studies. This referee realizes this may be a big “ask” – however if the authors have any insight on this issue, I’d encourage them to include it.

• Similarly, anything the authors can say about parameter sensitivity of these results would be of interest. I understand the computational/human cost may be prohibitive.

• In Figures 4 & 5 I am a little concerned about the difference between “Maximum gregarious locust density” and “Final peak density” – I believe the authors just took the maximum at the final time step of the simulation. Can you convince the reader this is the right measure? How do I know that the swarm didn’t gregarize and then somehow dissipate?

• Similarly, in Figures 4 & 5 and looking at the third figure in the fourth row of Figure 6, can the authors predict the peak density from the analysis in Figure 3? How do they compare?

• Line 322: The authors state: “as we found in certain cases the initial noise had an effect on whether a hopper band would form.” – yet Figures 4 & 5 only report one run (I believe). How did you report data in cases where sometimes you have a hopper band form and sometimes not?

• Figure 6: It seems likely that the symmetry in the problem plays a role here – can the authors comment on the mechanism behind the symmetry breaking seen in the second figure of row 3 and the third figure of row 4?

• Line 364: Can you reconcile the statement: “This effect is visualised in Fig 6, as gregarious locusts aggregate in the center of the food mass and displace their solitarious counterparts” with the third figure of row 4 where it appears the food source is entirely in the solitary portion of the swarm?

• Line 374: “Analytical investigations of our model shows that a spatially uniform food source has a variety of effects has on locust behavior” – eliminate the second appearance of “has” perhaps?

• Line 383: “Using this relationship we then found that our model also has a theoretical **maximum** locust density for hopper band formation, and that the presence of food lowers both the required time and density of locusts for hopper band formation.” Should maximum be minimum here? If not . . . you lost me. Please explain.</x,\\rho_xx>

**Have all data underlying the figures and results presented in the manuscript been provided?**

Reviewer #1: Yes

Reviewer #2: Yes

Reviewer #3: None

PLOS authors have the option to publish the peer review history of their article (what does this mean?). If published, this will include your full peer review and any attached files.

Reviewer #1: No

Reviewer #2: No

Reviewer #3: No
---

## [Decision Letter · Decision Letter 1]

29 Apr 2021

Dear Mr Georgiou,

Thank you very much for submitting your manuscript "Modelling locust foraging: How and why food affects hopper band formation" for consideration at PLOS Computational Biology. As with all papers reviewed by the journal, your manuscript was reviewed by members of the editorial board and by an independent reviewer. The reviewer appreciated the attention to an important topic. Based on the reviews, we are likely to accept this manuscript for publication, providing that you modify the manuscript according to the review recommendations.

Sincerely,

Jason Papin

Editor-in-Chief

PLOS Computational Biology

Feilim Mac Gabhann

Editor-in-Chief

PLOS Computational Biology

[LINK]

Reviewer's Responses to Questions

**Comments to the Authors:**

Reviewer #1: Second report on “Modelling locust foraging: How and why food affects hopper band formation”. The manuscript has been improved considerably. I recommend it is accepted to PLoS Comput Biol. I have two comments related to the presentation of the main conclusions.

1. As the authors explained in their reply letter, the model, which does not consider alignment, is not directly relevant to band formation. Instead, the model addresses aggregation, which is a required step before band formation. For this reason, I find that the repeated reference to band formation, including in the title, is confusing.

2. The interpretation of foraging as the “per capita contact with food” within a single food patch is not the standard definition of foraging efficiency. As far as I know, efficient foraging typically refers to the probability or rate of finding a patch. For example, from Wikipedia: “The marginal value theorem (MVT) is an optimality model that usually describes the behavior of an optimally foraging individual in a system where resources (often food) are located in discrete patches separated by areas with no resources. Due to the resource-free space, animals must spend time traveling between patches.” Therefore, the authors claim that “there exists a foraging advantage to being gregarious” is misleading.

Overall, my impression is that the main results are misrepresented.

**Have all data underlying the figures and results presented in the manuscript been provided?**

Reviewer #1: Yes

PLOS authors have the option to publish the peer review history of their article (what does this mean?). If published, this will include your full peer review and any attached files.

Reviewer #1: No

Figure Files:

Data Requirements:

Reproducibility:

References:

---

## [Editor Report · Decision Letter 2]

10 Jun 2021

Dear Mr Georgiou,

We are pleased to inform you that your manuscript 'Modelling locust foraging: How and why food affects group formation' has been provisionally accepted for publication in PLOS Computational Biology.

Best regards,

Iain Couzin

Guest Editor

PLOS Computational Biology

Stefano Allesina

Deputy Editor

PLOS Computational Biology

---

## [Editor Report · Acceptance letter]

28 Jun 2021

PCOMPBIOL-D-20-01682R2 

Modelling locust foraging: How and why food affects group formation

Dear Dr Georgiou,

I am pleased to inform you that your manuscript has been formally accepted for publication in PLOS Computational Biology. Your manuscript is now with our production department and you will be notified of the publication date in due course.

With kind regards,

Katalin Szabo
